# Identification of the Response-Related Biomarker of Bimonthly Hepatic Arterial Infusion Chemotherapy

**DOI:** 10.3390/jcm10040629

**Published:** 2021-02-07

**Authors:** Kei Moriya, Tadashi Namisaki, Hiroaki Takaya, Kosuke Kaji, Hideto Kawaratani, Naotaka Shimozato, Yasuhiko Sawada, Akitoshi Douhara, Shinya Sato, Masanori Furukawa, Koh Kitagawa, Takemi Akahane, Hitoshi Yoshiji

**Affiliations:** Department of Gastroenterology and Hepatology, Nara Medical University, Nara 634-8521, Japan; tadashin@naramed-u.ac.jp (T.N.); htky@naramed-u.ac.jp (H.T.); kajik@naramed-u.ac.jp (K.K.); kawara@naramed-u.ac.jp (H.K.); shimozato@naramed-u.ac.jp (N.S.); yasuhiko@naramed-u.ac.jp (Y.S.); aki-do@hotmail.co.jp (A.D.); shinyasato@naramed-u.ac.jp (S.S.); furukawa@naramed-u.ac.jp (M.F.); kitagawa@naramed-u.ac.jp (K.K.); stakemi@naramed-u.ac.jp (T.A.); yoshijih@naramed-u.ac.jp (H.Y.)

**Keywords:** hepatocellular carcinoma, hepatic arterial infusion chemotherapy, molecularly targeted therapies, hepatic functional reserve, des-gamma-carboxy prothrombin

## Abstract

Despite the availability of molecularly targeted agents for advanced hepatocellular carcinoma (aHCC), these are limited to compensated cirrhotic patients, and concerns about decreased hepatic functional reserve (HFR) and unknown adverse events, which may affect long-term survival, remain unaddressed. In this study, we enrolled 96 aHCC patients treated with bimonthly hepatic arterial infusion chemotherapy (B-HAIC) with cisplatin or sorafenib monotherapy (oral sorafenib 400 mg twice daily) not only to demonstrate its efficacy and significance but also to indicate preferable candidates by setting a response-related biomarker. Differences in treatment had no significant effect on overall survival (OS). The response rate in patients treated with B-HAIC was relatively higher than those treated with sorafenib. HFR was well maintained over the treatment course with B-HAIC, while it was significantly impaired with sorafenib. By employing multivariate analysis, we found negative trends between progression-free survival (PFS) periods and serum levels of alpha fetoprotein as well as des-gamma-carboxy prothrombin (DCP). In addition, a logistic regression analysis of the relationship between serum DCP levels and PFS periods over 420 days (14 months) showed that the PFS periods of patients with higher DCP was significantly shorter than those of patients with lower DCP (*p* = 0.02). Subsequently, the present study demonstrated the efficacy and safety of B-HAIC and identified a predictor of unpreferable patients. Based on these results, B-HAIC might be an alternative treatment after the implementation of new molecularly targeted therapies.

## 1. Introduction

Improvements in various diagnostic imaging techniques and significant advances in endoscopic techniques over the last few decades have led to improved treatment outcomes for cancers of the gastrointestinal tract, especially gastric and colorectal cancers, in terms of physical treatment burden and long-term prognosis. However, the five-year survival rates for cancers such as liver and pancreatic cancers remain low, representing intractable tumors [1].

Advanced hepatocellular carcinoma (aHCC), such as those associated with vascular invasion and distant metastases, is classed as grade C in the Barcelona Clinic Liver Cancer classification that is adopted in the European Association for the Study of the Liver (EASL) and American Association for the Study of Liver Diseases (AASLD) practice guidelines [2,3]. Molecularly targeted therapies are the recommended treatment for aHCC, and sorafenib is the only drug that has shown therapeutic efficacy for over a decade [4]. Lenvatinib recently became available in clinical practice after the REFLECT study demonstrated its noninferiority to sorafenib in 2018 [5]. Lenvatinib also demonstrated a more favorable outcome than transcatheter arterial chemoembolization (TACE) in patients with large or multinodular intermediate-stage HCC who did not benefit from TACE [6] and showed potential benefits for aHCC patients using second-line or later therapies and a high burden of intrahepatic lesions [7]. Based on these findings, the latest version of the HCC treatment practice guidelines recommends lenvatinib in addition to sorafenib for aHCC [2,3]. Recently, the combined used of atezolizumab, an immune checkpoint inhibitor (programmed cell death receptor ligand 1 antibody), and bevacizumab, a vascular endothelial growth factor receptor antibody product, showed greater overall survival (OS) compared with sorafenib monotherapy treatment [8]. These findings may represent a change in future treatment for aHCC. Although these two combination therapies are administered repeatedly to patients via intravenous infusion at three-week intervals, they are not recommended for use in patients with autoimmune diseases, which have increased in recent years, or those with a history of chronic or recurrent autoimmune diseases. Furthermore, as with existing molecularly targeted therapies, administration is limited to patients with compensated liver cirrhosis, and from a health economic standpoint, the cost of a single dose of the drug alone is very high, exceeding $10,000. Thus, we have entered an era in which multiple therapeutic agents for the treatment of aHCC have emerged with evidence. Furthermore, when the first-line of treatment fails, the choice of the second-line of treatment becomes a major issue. In order to implement these treatments, it is important to ensure good hepatic functional reserve (HFR) in patients. Low albumin levels have been reported to predict the long-term prognosis of patients with liver disease, including those with cirrhosis, as a decline in HFR often begins with decreased serum albumin levels [9].

We previously showed that bimonthly hepatic arterial infusion chemotherapy (B-HAIC) with one-shot cisplatin infusion at eight-week intervals could be an alternative treatment for TACE-refractory aHCC in both compensated and decompensated cirrhotic patients with aHCC treated in our hospital [10,11]. However, the preferable candidate of B-HAIC has not been clarified. In the present study, in addition to increasing the number of patients analyzed, we also summarized the outcomes of B-HAIC and sorafenib treatment over a period of up to seven years of follow-up and performed a detailed analysis of patient tumor background factors. Furthermore, it is generally known that des-gamma-carboxy prothrombin (DCP) is an abnormal form of the coagulation protein and is considered a complementary biomarker to alpha fetoprotein (AFP) for assessing the risk of developing HCC. By using these serum protein levels, we successfully demonstrated the response-related biomarker of B-HAIC by which its concerns and treatment limits could be predicted. Subsequently, the ideal post-treatment method for combination therapy with immune checkpoint inhibitors, which is expected to be widely used in the near future, is suggested.

## 2. Patients and Methods

### 2.1. Patients

The medical records of 103 chemonaïve patients refractory to TACE or those with distinct extrahepatic metastasis were reviewed to evaluate the efficacy and safety of B-HAIC and molecularly targeted agents for cirrhotic patients with aHCC. Eligible patients were admitted to the Nara Medical University Hospital between January 2009 and June 2015 and were enrolled into this retrospective study. Seven of the 103 patients were excluded due to death as a result of acute disease progression within a month of treatment initiation with B-HAIC or sorafenib. The dose intensity in these patients was extremely low, and it was actually hard to expect their therapeutic effects. As shown in Figure 1, 96 patients with advanced HCC were finally enrolled in this study.

### 2.2. Definition of TACE Failure

The preceding TACE sessions were performed using an emulsion containing anticancer agents and lipiodol followed by the application of gelatin sponge particles. The TACE refractory patients were diagnosed as per the guidelines of the Japan Society of Hepatology and the Liver Cancer Study Group of Japan [12,13].

### 2.3. Molecularly Targeted Therapeutics

Patients in the sorafenib group were treated using a maximum daily dose of 800 mg (oral sorafenib 400 mg twice daily). Each starting dose of sorafenib was tapered immediately if required (mainly due to adverse effects) and continued as long as tolerated without tumor progression. Dermatological examinations were performed routinely prior to starting treatment to minimize unfavorable adverse events, such as hand–foot syndrome.

### 2.4. B-HAIC Treatment and Assessment

B-HAIC treatment was performed as previously described [10,11]. Briefly, 65 mg/m^2^ cisplatin was administered intra-arterially over 30 min via a catheter inserted into the hepatic artery every eight weeks, for up to six courses, until either disease progression or unacceptable adverse events occurred. Patients exhibiting a desirable response to the six courses of B-HAIC were then sequentially treated with an implanted 5-fluorouracil reservoir. This treatment was continued until tolerated by the patient without disease progression or the occurrence of serious adverse effects. Dynamic enhanced computed tomography or magnetic resonance imaging was performed every four to eight weeks to confirm the antitumor effects of the treatment as per the modified Response Evaluation Criteria in Solid Tumors [14].

The choice of treatment in each case was determined by the physician team in charge based on the size, number, and stage of the cancer as well as hepatic functional reserve and renal function.

### 2.5. Statistical Analyses

Chi-square test and Mann–Whitney *U* test were used to compare the patient characteristics and antitumor effects between the groups. OS was calculated using the Kaplan–Meier method and between-group differences were compared using the log-rank test as well as the Wilcoxon test. Multivariate analysis and logistic regression analysis were used to show the relationship between progression-free survival (PFS) periods and serum markers. All *p-*values < 0.05 were considered statistically significant. Statistical analyses were performed using JMP version 14.3 software (SAS Institute Inc., Cary, NC, USA).

### 2.6. Ethics

Informed consent was obtained from all patients prior to treatment initiation. The study was approved by the Ethics Committee of the Nara Medical University Hospital (approval #000522 and #001490) and was conducted according to the ethical principles in the Japanese ethics guideline for epidemiological research [15].

## 3. Results

### 3.1. Patient Characteristics

Among 417 HCC patients seen at our hospital between January 2009 and June 2015, 103 patients with aHCC and refractory to TACE, adequately assessed by radiological imaging, and with sufficient blood test results were included in the study. Seven patients were excluded due to death within a month of initiation of B-HAIC or sorafenib. We analyzed the medical records of 48 patients with aHCC treated with B-HAIC and 48 patients treated with sorafenib. Patients treated with B-HAIC were then divided into two different groups according to their HFR for subanalyses (Figure 1). The clinical profiles of patients with aHCC are presented in Table 1, Appendix A. Patients with aHCC in the sorafenib (*n* = 48) and B-HAIC (*n* = 48) groups were comparable in terms of age, sex, preceding treatment ratio, tumor markers such as alpha fetoprotein (AFP) and DCP, and number of intrahepatic HCCs (Appendix A). On the other hand, cases classified as clinical stage III were most frequently in the B-HAIC group, whereas stage IV cases were most commonly found in the sorafenib group. Two cases with extrahepatic metastasis were included in the B-HAIC group.

### 3.2. Efficacy and Adverse Events

The best clinical response and OS rates for each group included in this retrospective observational study are shown in Figure 2A,B and Appendix A. The disease control rate (sorafenib, 58% and B-HAIC, 69%) was not significantly different between the groups, while the efficacy rate in the B-HAIC group was more than double that of the sorafenib group (27% vs. 10%, *p* = 0.065). As shown in Appendix A, the calculated OS period of the sorafenib group was similar to that of the B-HAIC Child A group (*p* = 0.378). On the other hand, the calculated OS period of the B-HAIC Child A group was significantly longer than that of the B-HAIC Child B group (*p* = 0.037) (Appendix A). It is worth noting that HFR was significantly impaired in patients in the sorafenib group according to the time course of treatment, whereas HFR did not change significantly during the treatment period in patients in the B-HAIC group (Figure 2C and Appendix A). No detrimental effects on renal function were observed in patients in either group (Figure 2D and Appendix A). The main reason for discontinuing treatment in the B-HAIC group was disease progression (41/48 cases (85.4%)). In contrast, 39.6% (19/48 cases) of the sorafenib group discontinued treatment due to other reasons, such as physical adverse events (Figure 2E and Appendix A). Patients in the B-HAIC group exhibited a similar rate of additional chemotherapy, such as 5-fluorouracil-based regimens and/or sorafenib, after the end of the B-HAIC treatments compared with the sorafenib group (Figure 2F and Appendix A). The best clinical response of additional treatments is shown in Appendix A. The disease control rate (sorafenib, 78% and B-HAIC, 75%) was similar to each other.

### 3.3. Relationship between Tumor Factors and Overall Survival Period

The number of intrahepatic HCCs present in the liver of each patient was measured to elucidate the relationship between clinical factors and OS period. The degree of tumor progression (“T factor”) was also determined according to the tumor/node/metastasis (TNM) classification system advocated by the American Joint Committee on Cancer [16]. The combination of intravascular invasion as well as extrahepatic metastasis was also examined. As shown in Appendix A, the number of HCCs did not affect the OS period between the B-HAIC Child A and sorafenib groups, whereas the OS between the B-HAIC Child A and B-HAIC Child B groups differed significantly (*p* = 0.012). In patients with T3 degree of HCC progression, the OS period of the B-AHIC Child A and sorafenib groups demonstrated an obviously elongated tendency compared with that of the B-HAIC Child B group (*p* = 0.055), although this difference was not recognized in patients with T4 degree of HCC progression (Appendix A). In terms of intravascular invasion (Appendix A), the OS was similar among the groups (B-HAIC Child A/B and sorafenib groups) (*p* = 0.878). The OS of the sorafenib groups with or without extrahepatic metastasis were not significantly different. On the other hand, the OS of the B-HAIC Child A group without extrahepatic metastasis was longer than that of the sorafenib group without metastasis during the three-year period after the initiation of each treatment, although the two super responders to sorafenib did not show this difference (Appendix A).

### 3.4. Use of Tumor Markers to Predict PFS Periods

We evaluated the relationship between PFS periods and serum tumor marker levels as well as biochemical indexes, including albumin, total bilirubin, and Child–Pugh scores. In the patients treated with B-HAIC, multivariate analysis revealed that the correlation between PFS and DCP as well as AFP was the strongest in these parameters (Figure 3A–E and Appendix A). In addition, a logistic regression analysis of the relationship between serum DCP levels and PFS periods over 420 days (14 months) showed that the PFS periods of patients with higher DCP was significantly shorter than those of patients with lower DCP (*p* = 0.0212). On the other hand, there was no obvious association between PFS periods and serum tumor marker levels in patients treated with sorafenib.

## 4. Discussion

Systemic therapy for HCC has changed remarkably since the introduction of the molecularly targeted agent, sorafenib, in 2007 [17]. However, in the following decade, many first-line and second-line agents that were developed failed to meet their primary endpoints in clinical trials [18]. On the other hand, four agents (regorafenib, lenvatinib, cabozantinib, and ramucirumab) successfully emerged from clinical trials in succession in 2017 and 2018 and became available or are underway in clinical practice [19]. In 2020, combination therapy using atezolizumab and bevacizumab was demonstrated to be significantly dominant agents compared with sorafenib monotherapy in randomized controlled trials [8]. Phase 3 clinical trials of immune checkpoint inhibitors and combination therapy of molecular targeted agents and immune checkpoint inhibitors are also underway [19]. This indicates an expected change in the treatment paradigm for HCC from early to advanced stage.

The prognosis of HCC patients is relatively poor as it is usually detected in its advanced stage as well as in patients with liver cirrhosis [1]. Various treatment approaches for HCC have been examined to improve the life expectancy of HCC patients. We investigated the efficacy of B-HAIC treatment for patients with aHCC up to now [10,11]. The advantages of one-shot intravenous infusion therapy include the short procedure time on the day (despite the requirement for a short hospital stay) and modest post-treatment complications compared with TACE, which is the standard treatment for HCC and is associated with frequent occurrence of postembolization syndrome consisting of transient fever, abdominal pain, and elevated transaminases as well as a number of severe adverse events, such as renal failure, gastroduodenal ulcerations, ascites, encephalopathy, and transient liver failure [20,21,22]. Thus, treatment is less invasive and has a less negative impact on HFR, which is important for long-term prognosis.

Using tumor-associated proteins with short half-lives, such as AFP and DCP [23], enables the post-treatment efficacy to be determined within two weeks [24]. On the other hand, patients positive for hepatitis C virus antibody, the main cause of liver cirrhosis in Japan, are unevenly distributed among the elderly population, and patients with liver cirrhosis are generally older than those without the condition. Therefore, despite these advantages, repeated cisplatin infusion at four-week intervals, which is the conventional method, is burdensome in terms of anorexia and general malaise as well as hematologic toxicity. Therefore, we chose the B-HAIC regimen, which comprises cisplatin administered at eight-week intervals, in cases in which the disease was under control after a rapid evaluation of its efficacy, as previously reported [24]. Although some patients may not respond to this treatment, they can be treated afterwards with other therapies, such as molecularly targeted agents, without compromising HFR. It is always important to extend the OS of patients by administering multiple therapies, including local treatment, as required.

The present study demonstrated two main findings. First, B-HAIC is a HFR-friendly treatment for patients with aHCC, according to the assessment of efficacy of B-HAIC on median survival time, and may be expected to show a comparable or superior outcome to sorafenib. In other words, B-HAIC, which depends on the antitumor effect of the cytotoxic agent, cisplatin, does not impair HFR. However, even in cases with a good response, changes in the regimen are necessary to avoid accumulated toxicity, such as neuropathy caused by cisplatin, and the regimen eventually leads to a fatal course of events, such as tumor recurrence. On the other hand, although the efficacy rate of sorafenib was almost lower than that of B-HAIC (Appendix A), there was a case of super response to these molecularly targeted agents (Appendix A), which is expected to have a long-term survival of more than several years [25]. Therefore, hepatologists must predict which HCC patients could successfully respond well to molecularly targeted therapies prior to treatment.

Second, the results of this clinical study suggest that stratification of pretreatment tumor markers may be predictive of long-term outcomes in patients with aHCC. As previously reported, HCC patients with high levels of both AFP and DCP tumor markers are assumed to have a poor prognosis. In other words, high levels of both tumor markers are an independent risk factor associated with tumor recurrence [26]. Saito et al. reported that high values of DCP prior to TACE were associated with long-term deterioration of liver function [27]. The present study found a correlation between the PFS periods and pretreatment levels of DCP in patients treated with B-HAIC. This fact might also suggest the possibility of DCP as a prognostic predictor in patients with advanced HCC. Recently, we reported that the imbalance of ADAMTS13 activity and von Willebrand factor (VWF) antigen might be a predicting factor of an initial treatment response of B-HAIC [28]. By combining these factors with serum DCP level before treatment, more accurate prediction of treatment effect may be available.

The present study has some limitations. First, although the study performed long-term observation, it was a single-center, retrospective observational study with a limited number of patients. Second, the criteria for choosing between B-HAIC and sorafenib were left to the decision of the hepatologists at each clinical group. Third, the physicians in charge of the study worked individually to choose the therapeutic method and determine whether or not post-treatment therapy should be administered. On the other hand, our hospital is a flagship hospital in its region, and there are no other medical institutions in the vicinity that can treat aHCC. Therefore, our hospital is an ideal environment for treating almost all patients with aHCC until the end of their lives and is able to maintain a constant standard of care for HCC patients under observation. However, regardless of these limitations, cisplatin remains an important antitumor agent against various malignant neoplasms and the significance of the use of B-HAIC, which allows for even safer and more effective use of cisplatin than previously, should be fully appreciated.

In conclusion, the present study demonstrated the efficacy and safety of B-HAIC and identified a predictor of poor OS with this treatment. In addition, the efficacy of B-HAIC as a second-line or third-line treatment after the implementation of new molecularly targeted therapies, which are widely selected and implemented, represents a good alternative treatment.

## 5. Study Highlights

We recruited 96 aHCC cases treated with B-HAIC or sorafenib monotherapy to demonstrate their efficacy and significance. The OS in both the B-HAIC (*n* = 48) and sorafenib treatment groups (*n* = 48) was comparable. HFR in the B-HAIC group was maintained well, while that in the sorafenib group was significantly decreased. The B-HAIC group showed a significantly lower likelihood of long-term PFS in patients with high DCP levels. The outcome of aHCC patients treated with B-HAIC was comparable to those treated with sorafenib. However, the long-term survival rate with B-HAIC may be lower than with sorafenib monotherapy in patients with high DCP levels prior to treatment initiation, suggesting that molecularly targeted therapies should be the preferred choice for such patients.

## Figures and Tables

**Figure 1 jcm-10-00629-f001:**
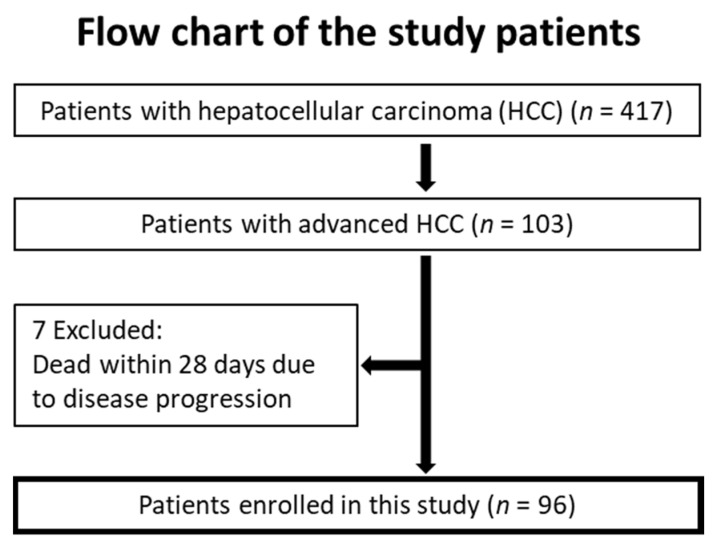
Patient flow chart. Data from 417 patients with hepatocellular carcinoma (HCC) were initially assessed. Among these patients, 103 had advanced HCC (aHCC) and seven were excluded due to death from acute disease progression within four weeks of treatment initiation. A total of 96 patients with advanced HCC were finally enrolled in this study.

**Figure 2 jcm-10-00629-f002:**
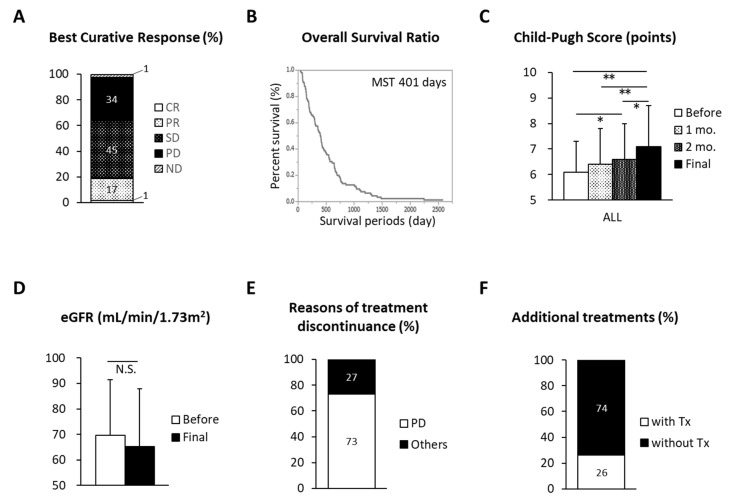
Chemopreventive effects on advanced HCC and adverse effects in this study. (**A**) Best curative response of the HCC patients in this study. CR, complete response; PR, partial response; SD, stable disease; PD, progressive disease; ND, not determined. (**B**) Kaplan–Meier curves show the overall survival (OS) of patients (*n* = 96). (**C**,**D**) Changes in Child–Pugh scores and estimated glomerular filtration rate (eGFR) before and after the treatments. (**E**) Reasons of treatment discontinuance. (**F**) Ratio of additional treatments after the failure of initial treatment. * *p* < 0.05, ** *p* < 0.01.

**Figure 3 jcm-10-00629-f003:**
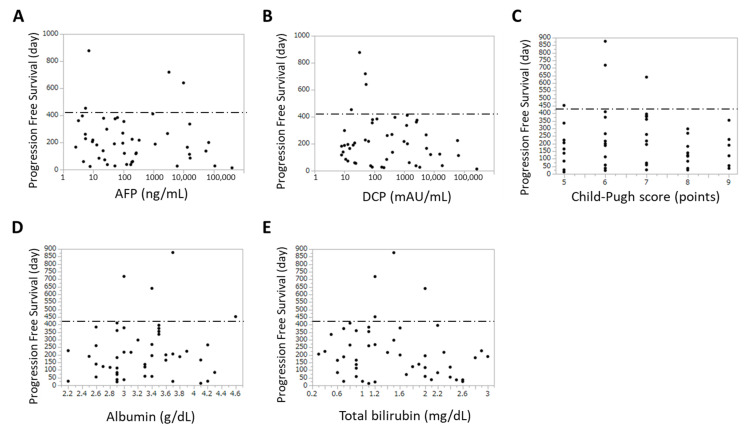
Co-relationship between progression-free survival (PFS) and the levels of pretreatment tumor markers as well as other biochemical indexes in patients treated with B-HAIC. (**A**) Co-relationship between PFS and alpha fetoprotein. (**B**) Co-relationship between PFS and des-gamma-carboxy prothrombin. (**C**) Co-relationship between PFS and Child-Pugh score. (**D**) Co-relationship between PFS and albumin. (**E**) Co-relationship between PFS and total bilirubin.

**Table 1 jcm-10-00629-t001:** Profiles of HCC patients with liver cirrhosis in this study (*n* = 96).

Age (Years, Median)	72.0 (43–88)
Sex (male/female)	78/18
Etiology (HBV/HCV/alcohol/others)	18/66/7/5
Child–Pugh classification (class A/class B)	70/26
Preceding treatments (yes/no)	90/6
AFP (ng/mL) (median (range))	99 (2.6–406,875)
DCP (mAU/mL) (median (range))	388 (8–268,747)
HCC numbers (1–3/4 and over)	24/72
Intravascular invasion (with/without)	25/71
Extrahepatic metastasis (with/without)	24/72
HCC clinical stage (II/III/IV)	15/36/45
T factor (T2/T3/T4)	16/51/29
HCC treatment (B-HAIC/sorafenib)	48/48

HBV, hepatitis B virus; HCV, hepatitis C virus; HCC, hepatocellular carcinoma; AFP, alpha-fetoprotein; DCP, des-gamma-carboxy prothrombin; B-HAIC, bimonthly hepatic arterial infusion chemotherapy.

## Data Availability

The datasets used and/or analyzed during the current study are available from the corresponding author on reasonable request.

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
