# Peer review of "Identification of the Response-Related Biomarker of Bimonthly Hepatic Arterial Infusion Chemotherapy"

_jcm, 2021, doi:10.3390/jcm10040629_

Round 1
Reviewer 1 Report
It’s a very useful clinic study.
The abstract is not clear, could the author reorganize it.
Did the author test some other serum biomarkers except AFP, such as GPC3. And how about the correlation among them? Meanwhile, did the author also test the serum of liver function, how about the AST, ALT, ALP between different group?
In the introduction part, since the author would like to show the diagnostic biomarker in the liver cancer treatment to predict the treatment outcome, could the author add some introduction of DCP used in the study in the part?
The statistics analysis part on DCP null hypothesis test is not clear, could the author modify it?
About table 1, since the author showed a difference between the two group patients, how about the different treatment responses between them, not only shown the analysis on the whole group shown in figure 2, supplementary figure 2( tumor factor and extrahepatic metastasis).
Author Response
Reviewer 1
It’s a very useful clinic study. The abstract is not clear, could the author reorganize it.
Thank you for your comments. According to your suggestion, we rewrote the abstract for making it clearer. Please see the abstract section, (the line 7 and 22). We also added some minor changes in the first half of the study highlight section. Please see the line 274 and 277.
Did the author test some other serum biomarkers except AFP, such as GPC3. And how about the correlation among them? Meanwhile, did the author also test the serum of liver function, how about the AST, ALT, ALP between different group?
Unfortunately, serum biomarker GPC3 has not been approved by national health insurance in our country and we have no data of it. In addition to AFP and des-gamma carboxy prothrombin (DCP), the Lens culinaris agglutinin-reactive fraction of AFP (L3-AFP) is the other approved tumor marker of HCC. However, the data of L3-AFP in this study were almost blank and could not induce the definite results.
Concerning to the levels of AST, ALT, and ALP, we had not collected these parameters in this study and could not show them in the current revise. I deeply apologize for this.
Instead, we would like to indicate the relations between PFS and some other biomarkers such as albumin, total bilirubin, and Child-Pugh score which reflects hepatic functional reserve in the patients treated with B-HAIC. These parameters had been successfully collected in advance (Please see the Figure 3). Subsequently, the correlation between PFS and DCP (as well as AFP) was the strongest in these parameters (Please see the supplementary Table 3). We added the following sentences “We evaluated the relationship between PFS periods and serum tumor marker levels as well as biochemical indexes including albumin, total bilirubin, and Child-Pugh scores. In the patients treated with B-HAIC, multivariate analysis revealed that the correlation between PFS and DCP as well as AFP was the strongest in these parameters (Fig. 3A-E and Supplementary Table 3).” in the result section, (Please see the line 180 and 183).
In the introduction part, since the author would like to show the diagnostic biomarker in the liver cancer treatment to predict the treatment outcome, could the author add some introduction of DCP used in the study in the part?
Thank you for your appropriate comment. We added the following sentences in the introduction section “it is generally known that des-gamma-carboxy prothrombin (DCP) is an abnormal form of the coagulation protein and is considered a complementary biomarker to alpha fetoprotein (AFP) for assessing the risk of developing HCC. By using these serum protein levels,”, (Please see the line 62 and 64).
The statistics analysis part on DCP null hypothesis test is not clear, could the author modify it?
We appreciate your comment. According to your suggestion, we modified the statistics analysis part on DCP. We deleted the following sentences “We planned a comparative study of progression-free survival (PFS) periods classified by the levels of serum tumor marker in the two independent groups treated with a different approach. Cases were initially treated with B-HAIC and controls were with sorafenib. The ratio of cases to controls was 1. Our preliminary data indicated that the probability of exposure (DCP≥60 mAU/ml and PFS≥12 months) in the controls was 0.25. If the true probability of exposure in the cases is 0.04, we will need to study 43 patients treated with sorafenib and 43 patients with B-HAIC to be able to reject the null hypothesis that the exposure rates for case and controls are equal with probability (power) 0.8. The Type I error probability associated with this test of this null hypothesis is 0.05. We will use Chi-squared statistic to evaluate this null hypothesis.” in the patients and methods section and we added the following sentences “Multivariate analysis and logistic regression analysis were used to show the relationship between progression-free survival (PFS) periods and serum markers.” in the same section, (Please see the line 107 and 109).
We also deleted the following sentences “We evaluated the relationship between serum tumor marker levels and PFS periods in the B-HAIC and sorafenib groups (Fig. 3A–D). There was no obvious association between serum AFP levels and PFS periods in the B-HAIC and sorafenib groups. On the other hand, there was only one case in the B-HAIC group with serum DCP levels >60 mAU/mL (1.5 times above the upper limit of the normal range) and its PFS periods exceeded 400 days (almost one year). Pearson’s Chi-square test revealed a significant difference between the four subgroups categorized by DCP and PFS (p = 0.038), although the AFP levels (ng/mL) in the patients with higher DCP (≥60 mAU/mL, n = 30) was not significantly different compared with patients with lower DCP (<60 mAU/mL, n = 18) (p = 0.270). Interestingly, six cases in the sorafenib group had serum DCP levels >60 mAU/mL and PFS period >400 days, and there was no significant difference between the four subgroups (p = 0.436).” in the result section and “The present study found no clear trend between pretreatment AFP and PFS with B-HAIC or sorafenib, except for a very weak negative correlation (Fig. 3A, B). However, when patients in both the sorafenib and B-HAIC groups were divided according to DCP levels into two groups (≥60 mAU/mL and <60 mAU/mL, which was 1.5 times above the upper normal limit), the percentage of ≥400 days PFS in the B-HAIC group was significantly lower than that of the population with DCP levels <60 mAU/mL (p = 0.038), indicating that high pretreatment DCP levels in the B-HAIC group did not indicate long-term PFS (Fig. 3C). A similar analysis of the sorafenib group showed no clear correlation between pretreatment DCP levels and long-term outcome in this group (p = 0.436) (Fig. 3D).” in the discussion section. Then, we added the following sentences in the results section “We evaluated the relationship between PFS periods and serum tumor marker levels as well as biochemical indexes including albumin, total bilirubin, and Child-Pugh scores. In the patients treated with B-HAIC, multivariate analysis revealed that the correlation between PFS and DCP as well as AFP was the strongest in these parameters (Fig. 3A-E and Supplementary Table 3). In addition, a logistic regression analysis of the relationship between serum DCP levels and PFS periods over 420 days (14 months) showed that the PFS periods of patients with higher DCP was significantly shorter than those of patients with lower DCP (P = 0.0212). On the other hand, there was no obvious association between PFS periods and serum tumor marker levels in the patients treated with sorafenib (data not shown).” and “The present study found a correlation between the PFS periods and pretreatment levels of DCP in patients treated with B-HAIC. This fact might also suggest the possibility of DCP as a prognostic predictor in patients with advanced HCC.” in the discussion section. Please see the line 187 and 187, 239 and 241.
About table 1, since the author showed a difference between the two group patients, how about the different treatment responses between them, not only shown the analysis on the whole group shown in figure 2, supplementary figure 2 (tumor factor and extrahepatic metastasis).
Thank you for important comments. We checked the best curative response of additional treatments in each group. There was no significant change between the groups (Please see the Supplementary Figure 1G). We added the following sentences in the result section “The best clinical response of additional treatments is shown in Supplementary Fig. 1G. The disease control rate (sorafenib, 78% and B-HAIC, 75%) was similar to each other.”, (Please see the line 153 and 154).
Reviewer 2 Report
This is a retrospective study about two treatments of TACE refractory patients with HCC.
My major concern of this study is the large number of patients excluded from the analysis. There are 7 patients excluded due to early progression. This makes the any survival analyses meaningless.
Further more failed the authors to describe why one group got B-HAIC and the other Sofafenib. This makes it difficult to interpretate the end results.
However the patient data are of value, but the focus of the publication should change to a report of a serie and not on a comparison.
Author Response
Reviewer 2
This is a retrospective study about two treatments of TACE refractory patients with HCC. My major concern of this study is the large number of patients excluded from the analysis. There are 7 patients excluded due to early progression. This makes the any survival analyses meaningless. Furthermore, failed the authors to describe why one group got B-HAIC and the other Sofafenib. This makes it difficult to interpretate the end results.
I really appreciate your comment. The reason why we excluded 5 patients treated with sorafenib in this study was that they had not been able to continue the sorafenib treatment for more than 2 weeks and their dose intensities should be extremely low. Concerning the excluded 2 patients treated with B-HAIC, they had huge HCC (> 15 cm) in the liver with vulnerable hepatic functional reserve and died promptly after treatment due to hepatorenal syndrome. Based on these facts, we decided to exclude 7 patients in this study. We added the following sentences in the patients and methods section “The dose intensity in these patients was extremely low and it was actually hard to expect their therapeutic effects.”, (Please see the line 76 and 78).
According to your valuable suggestion, we added the following sentences in the patients and methods section for clarifying the criteria of deciding treatment methods on each patient with HCC “The choice of treatment in each case was determined by the physician team in charge, based on the size, number, and stage of the cancer, as well as hepatic functional reserve and renal function.”, (Please see the line 102 and 103).
However, the patient data are of value, but the focus of the publication should change to a report of a serie and not on a comparison.
Thank you for your important comments. According to your suggestion, we changed the style of our article from a comparison into a series. Please see the newly constructed figures and tables, (Please see the Figure 1, Figure 2, Table 1, supplementary Figures, and supplementary Tables). We deleted the following sentences “As shown in Fig. 1, 48 patients treated with sorafenib with Child–Pugh class A HFR were enrolled to the sorafenib group and 48 patients treated with B-HAIC were enrolled to the B-HAIC group and divided into two different groups according to HFR grade. Patients with compensated cirrhosis were enrolled as the B-HAIC Child A group and those with decompensated cirrhosis were enrolled as the B-HAIC Child B group.” and added the following sentences “As shown in the Fig 1, 96 patients with advanced HCC were finally enrolled in this study”, in the patients and methods section, (Please see the line 78 and 79).
Round 2
Reviewer 2 Report
The manuscript is much improved. To my opinion all issues of the previous versions are solved.
Author Response
I really appreciate your reviewing of my article.